# Differentially Private Approximate Near Neighbor Counting in High Dimensions

**Alexandr Andoni**
Columbia University
andoni@cs.columbia.edu

**Piotr Indyk**
MIT
indyk@mit.edu

**Sepideh Mahabadi**
Microsoft Research
smahabadi@microsoft.com

**Shyam Narayanan**
MIT
shyamsn@mit.edu

## Abstract

Range counting (e.g., counting the number of data points falling into a given query ball) under differential privacy has been studied extensively. However, the current algorithms for this problem are subject to the following dichotomy. One class of algorithms suffers from an additive error that is a fixed polynomial in the number of points. Another class of algorithms allows for polylogarithmic additive error, but the error grows exponentially in the dimension. To achieve the latter, the problem is relaxed to allow a "fuzzy" definition of the range boundary, e.g., a count of the points in a ball of radius $r$ might also include points in a ball of radius $cr$ for some $c > 1$. In this paper we present an efficient algorithm that offers a sweet spot between these two classes. The algorithm has an additive error that is an arbitrary small power of the data set size, depending on how fuzzy the range boundary is, as well as a small $(1 + o(1))$ multiplicative error. Crucially, the amount of noise added has no dependence on the dimension. Our algorithm introduces a variant of Locality-Sensitive Hashing, utilizing it in a novel manner.

## 1  Introduction

Differential Privacy (DP) [DMNS06, DKM$^+$06, DMNS16] is a widely used tool for preserving the privacy of sensitive personal information. It allows a data structure to provide approximate answers to queries about the data it holds, while ensuring that the removal or addition of a single database entry does not significantly affect the outcome of any analysis. The latter guarantee is accomplished by adding some amount of noise to the answers, so that the data cannot be "reverse-engineered" from the answers to the queries. See Definition 2.1 for the formal setup. The notion has been deployed in many important scenarios in industry [DKY17, EPK14, BEM$^+$17, Sha14, G$^+$16] as well as the U.S. census [Abo18].

One of the key data analysis problems studied under differential privacy is *range counting*. Here, the goal is to construct a data structure that answers queries about the total number of data items in a database satisfying a given property. Formally, given a multiset $X$ of $n$ elements from a universe $\mathcal{U}$, the goal of *range counting* queries is to report the number of points within a given range $Q \in \mathcal{Q}$ from a prespecified class of query ranges $\mathcal{Q}$. For example, this could correspond to retrieving the number of users in a particular geographic location, or a number of patients with given symptoms.

Differentially private range counting has been studied extensively, for many classes of queries. For example, the case of $\mathcal{Q} = \mathcal{U}$, i.e., when the queries ask for the number of occurrences of a given element in the database, this correspond to the well-known *histogram problem* [HKR12, EPK14].

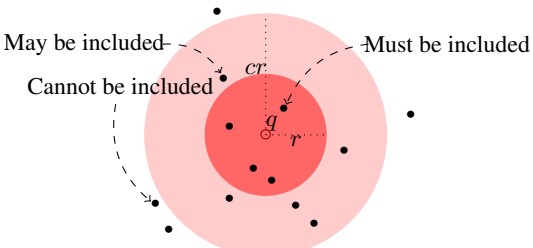

Figure 1: An illustration of approximate near neighbor query, with parameters $c$ and $r$.

More generally, a large body of research has focused on the case where universe $\mathcal{U}$ is the $d$-dimensional space $\mathbb{R}^d$ (or its discretization), and $\mathcal{Q}$ consists of natural geometric objects, like rectangles, half-planes or balls. The seminal work of [MN12] demonstrated a strong relationship between the amount of additive noise necessary to guarantee the privacy, and the *discrepancy* of the space of queries $\mathcal{Q}$. Unfortunately, except for a few classes of queries such as axis-parallel rectangles, most natural geometric objects have discrepancy that is polynomial in the number of data points $n$. This means that, in order to guarantee privacy, it is necessary to distort the counts by an additive term $\Theta(n^\rho)$, for a *fixed* constant $\rho > 0$ that depends on $\mathcal{Q}$.

To alleviate this issue, [HY21] introduced differentially private data structures for *approximate* range queries. For simplicity, we recap their notion for the special case of ball queries that we consider in the rest of the paper. We assume that $\mathcal{U}$ is a metric space equipped with a distance function $\mathrm{dist}$. For a radius parameter $r$, we define the query family $\mathcal{Q} = \{B(q, r) : q \in \mathcal{U}\}$, where $B(q, r) = \{p \in \mathcal{U} : \mathrm{dist}(p, q) \leq r\}$ is the ball of radius $r$ around $q$. In other words, in the *exact* problem, the goal is to count the number of data points that are "near" the given query point. In turn, the *approximate* problem is parameterized by an approximation factor $c \geq 1$, and the algorithm is allowed to report any number between $|X \cap B(q, r)|$ and $|X \cap B(q, cr)|$. Figure 1 illustrates this notion. We will refer to the approximate version as the $(c, r)$-near neighbor counting problem.

The main result of [HY21] shows that, if one allows approximate queries as defined above, then one can construct a data structure that guarantees privacy while adding much less noise. Specifically, if $\mathcal{U} = [u]^d$ (i.e., the universe is the discrete $d$-dimensional space) for some fixed $d = O(1)$ and $\mathrm{dist}(p, q) = \|p - q\|_s$ for some $\ell_s$ norm, then their algorithm achieves privacy while adding noise of magnitude roughly $(O(1/\alpha))^d \cdot \mathrm{polylog}(u + n)$, where $0 < \alpha < 1$ is a constant that depends on $c > 1$. This guarantees that the noise amount is only polylogarithmic, as long as the dimension $d$ is constant. However, the exponential dependence of the noise bound on the dimension precludes the applicability of this algorithm to the common case of high-dimensional data sets.

**Our contribution**   Our main result is an efficient data structure that can approximate the number of near neighbors to any query point over Euclidean ($\ell_2$) space, with differential privacy (see Preliminaries for the formal of definition of privacy). Specifically, for any fixed radius $r$, and any series of query points $q$, the data structure can privately answer $(c, r)$-near neighbor counting queries, where we are allowed a small additive error ($n^\rho$ for a small constant $\rho$) and multiplicative error $1 \pm o(1)$). Let $B_p(q, r)$ denote the $\ell_p$ ball of radius $r$ around $q$.

**Theorem 1.1.** *Fix a radius $r > 0$ and constant $c \geq 1$, and privacy parameters $1 > \epsilon, \delta > 0$. For any integer $n$ and dimension $d$, there exists a differentially private algorithm that, on a dataset $X \subset \mathbb{R}^d$ of at most $n$ points, creates a data structure that can answer $m$ (non-adaptive) queries and that satisfies the following properties.*

1. *(Privacy) The full output of the algorithm is $(\epsilon, \delta)$-differentially private.*

2. *(Accuracy) With high probability, for every non-adaptive query $q$, the output $\mathrm{ans}(q)$ is between*

$$(1 - o(1)) \cdot |X \cap B_2(q, r)| - n^{\rho + o(1)} \cdot \frac{\log(\log m/\delta) \cdot \log m}{\epsilon}$$

*and*

$$(1 + o(1)) \cdot |X \cap B_2(q, cr)| + n^{\rho + o(1)} \cdot \frac{\log(\log m/\delta) \cdot \log m}{\epsilon},$$

*where $\rho = \frac{4c^2}{(c^2+1)^2} = O(\frac{1}{c^2})$.*

3. *(Runtime) The preprocessing time is $O(n^{1+o(1)} \cdot d \cdot \log m)$, and the expected runtime for answering each query is $O(n^{\rho+o(1)} \cdot d \cdot \log m)$.*

The key two features of the above result are as follows. First, the amount of additive noise is controlled by the amount of allowed approximation: the exponent $\rho$ tends to $0$ as $c$ grows large. This makes it possible to trade the "proximity" error $c$ and the "privacy" error $n^\rho$ to achieve the best results. Second, the magnitude of added noise does not depend on the dimension; in particular, it avoids the exponential dependence of [HY21].

Finally, we remark that our algorithm can be extended to alternative/more general settings:

1. **$\ell_1$ metric:** If we considered the balls $B_1(q,r)$ rather than $B_2(q,r)$, we can still obtain a similar result, at the cost of having $\rho \approx \frac{4c}{(c+1)^2} = O(\frac{1}{c})$. This is based on the fact that the $\ell_1$ metric embeds near-isometrically into the squared $\ell_2$ metric.

2. **Pure differential privacy:** We can modify our algorithm to handle pure-DP queries (i.e., when $\epsilon = 0$), while still having good accuracy. In this case, however, the algorithm, while fast in expectation, has some small probability of being very slow.

3. **Large $\epsilon$:** Our results also hold even when $1 \leq \epsilon \leq n^{o(1)}$.

4. **Adaptive queries:** Our method also extends to adaptive queries, at the expense of the runtime and additive error multiplying by a factor of $\mathrm{poly}(d \log R/r)$, where $R$ is some promised bound on the radius of the dataset.

5. **$k$-NN queries:** Finally, we note that using similar approach to that of [HY21], one can get a data structure for finding an approximation to the distance of the query to its $k$-nearest neighbor. However, as opposed to [HY21], our data structure works only for a fixed value of $r$, and thus one needs to build separate data structures for various choices of $r$ leading to worse privacy guarantees which depend on the aspect ratio.

We discuss extensions 1–4 more formally in Appendix C.

A natural question is whether the tradeoff between approximation ratio $c$ and additive error $n^\rho$ is necessary, or whether one can obtain $c = O(1)$ and $\rho$ to be an arbitrarily small constant. While we are unable to prove such a lower bound in the Euclidean setting, we show that under the closely related $\ell_\infty$ norm for approximation ratio $c = 3 - o(1)$, one must have an additive error of $n^{\Omega(1)}$.

**Theorem 1.2.** *For sufficiently large $n$, there exists $d = C \log n$ for a large constant $C$, fixed constants $1 > \epsilon, \delta, \rho > 0$, and $n$ query points $Q = \{q_1, \ldots, q_n\}$, with the following property. For any (arbitrarily small) $\alpha > 0$ and any differentially private algorithm $\mathcal{A}$ that acts on a dataset $X \subset \mathbb{R}^d$ of at most $n$ points, and outputs $\{ans(q)\}_{q \in Q}$, must have*

$$\mathbb{P}\big(\forall q \in Q, \quad |B_\infty(q, 0.5)| - n^\rho \leq ans(q) \leq |B_\infty(q, 1.5 - \alpha)| + n^\rho\big) < \frac{2}{3}.$$

Theorem 1.2 provides some evidence that a tradeoff may be necessary even in the $\ell_2$ case: proving such a result is an interesting open problem. We defer the proof of Theorem 1.2 to Appendix B.

**Overview of the techniques.** Our high-level approach is similar to that of the low-dimensional approximate range query data structure of [HY21], and consists of the following steps. First, we design a partition of $\mathbb{R}^d$, and count the number of points that fall into each cell in the partition. Then we add Laplace noise to each cell to ensure the privacy of the counts. Finally, for each query $q$, we approximate the ball $B(q,r)$ using the cells in the partition, and report the sum of the (noisy) counts of the approximating cells.

However, applying this approach as described requires using an exponential (in $d$) number of partition cells to approximate a query ball $B(q,r)$, leading to the exponential error bound. To overcome this issue, we resort to *randomized* space partitions obtained using Locality-Sensitive Hashing (LSH) techniques [IM98]. These partitions have the property that, for any two points $p$ and $q$, the probability that both of them belong in the same partition cell depends on the distance $\|p - q\|_2$. As shown in [Pan06] in the context of (non-private) approximate near neighbor search, enumerating a bounded

number ($n^\rho$) of cells in such partition makes it possible to report each data point from the range $B(q, r)$ with probability at least $1/\text{poly}(\log n)$. To amplify the probability of correctness, the process is repeated $O(\text{poly}\log n)$ times, using a different random partition in each repetition.

Unfortunately, the aforementioned approach does not appear to yield an efficient algorithm for *approximately counting* the number of points in $B(q, r)$. The main issue is that, as indicated above, when enumerating the $n^\rho$ cells corresponding to a query's ball, we are guaranteed to include only a $1/\text{poly}(\log n)$ fraction of points, leading to a large approximation factor. The standard way to deal with this — using multiple partitions — yields inconsistent counts which are difficult to aggregate. A number of follow-up works showed improved bounds on exponent $\rho$, but all of them similarly have a guarantee of identifying only a $\leq 1/\text{poly}(\log n)$ fraction of points [LJW+07, AI06, Kap15, Chr17, ALRW17]. To overcome these obstacles, we show how we can adapt the algorithm from [ALRW17], which yield an approximate near neighbor data structure with the best known tradeoff between space and time. However, their randomized algorithm does not yield partitions (technically, they construct so-called *locality-sensitive filters*, which are not partitions).

Our main technical contribution is two-fold. First, we show it is possible to "force" the algorithm of [ALRW17] to construct proper partitions. The quality of those partitions is comparable to the original structures, at the price of a more involved analysis. We then use those partitions to compute the counts and approximate the query ball as outlined at the beginning of this section. Second, we show that it is also possible to modify the algorithm so that the probability of including each qualifying point into the count is at least $1 - o(1)$. This is of separate interest, as this yields the most efficient algorithm for approximate nearest neighbor search with space $O(nd)$, improving over [Kap15].

## 1.1   Related work

The problem of range counting when the query class $\mathcal{Q}$ is arbitrary, can be solved with an optimal error bound of $\sqrt{n} \cdot \text{poly}(1/\epsilon, \log \frac{|\mathcal{Q}| \cdot |\mathcal{U}|}{\delta})$[HR10]. In the setting of approximate near-neighbor queries, the size of $\mathcal{Q}$ and $\mathcal{U}$ can be thought of as at most exponential in the dimension $d$, by using a standard $\epsilon$-net technique. Hence, their error is in fact smaller than ours when $\frac{4c^2}{(c^2+1)^2} \geq \frac{1}{2}$, or equivalently, when $c \leq 1 + \sqrt{2}$. However, whenever an approximation of $c > 1 + \sqrt{2}$ is acceptable, our error is smaller than [HR10]. Moreover, our approach is computationally efficient, whereas their method has runtime linear in the size of $\mathcal{U}$, which is very inefficient.

When the query class is restricted to points in $\mathcal{U}$, i.e., histogram queries, then the best achievable error is $O(\frac{1}{\epsilon} \cdot \min\{\log |\mathcal{U}|, \log(1/\delta)\})$[Vad17]. When the points are in 1-dimensional Euclidean space taking values in $[u]$, and $\mathcal{Q}$ is a set of intervals, [DNPR10] got an $(\epsilon, 0)$-DP algorithm with error $O(\frac{1}{\epsilon} \cdot \log^{1.5} u)$ and a lower bound of $\Omega(\frac{\log u}{\epsilon})$. When resorting to $(\epsilon, \delta)$-DP, [BNSV15] showed an algorithm with error $2^{(1+o(1))\log^* u} \log(1/\delta)/\epsilon$ and a lower bound of $\Omega(\log^* u \cdot \log(1/\delta)/\epsilon)$ for $e^{-\epsilon n/\log^* n} \leq \delta \leq 1/n^2$. The problem for the axis-parallel rectangles have been further studied [CSS11, DNRR15] where polylogarithmic error bounds were obtained.

More generally, [MN12, NTZ13] showed an equivalence between the error and the discrepancy of the range space $\mathcal{Q}$. Given that the discrepancy of many natural range spaces such as half-spaces, simplices, and spheres is $n^{\alpha(d)}$ where $\alpha(d) \in [1/4, 1/2]$ is a constant depending on $d$ [Mat99], it rules out such a polylogarithmic error for these range spaces. For non-convex ranges, the discrepancy already becomes $\sqrt{n}$ reaching the threshold for arbitrary query ranges.

There are a number of papers based on space decompositions that provide a DP data structure for range counting that perform relatively well in practice [CPS+12, HRMS09, LHMW14, QYL13a, QYL13b, XWG10, ZXX16] but perform poorly on high-discrepancy point sets.

Finally, we note that several prior works used LSH for differentially private kernel estimation [CS20, WNM23]. However, those works do not seem to be applicable to ball range queries, which correspond to uniform kernels.

## 2 Preliminaries

**Range Counting:** In the range counting problems, given a point set $X \subseteq \mathcal{U}$, where $\mathcal{U}$ is a universe, the goal is to construct a differentially private data structure that for any query $Q \subseteq \mathcal{U}$ from a certain query family $\mathcal{Q}$, one can compute $|Q \cap X|$.

Here we focus on Near Neighbor counting queries in high dimensions. In particular $\mathcal{U}$ is a metric space equipped with a distance function dist. Given a prespecified radius parameter $r$, the query family $\mathcal{Q} = \{B(q, r): q \in \mathcal{U}\}$, where $B(q, r) = \{p \in \mathcal{U}: \text{dist}(p, q) \leq r\}$ is the ball of radius $r$ around $q$. In other words, in this problem the goal is to count the number of *neighbors* of a given query point.

Similar to the work of [HY21], we consider the approximate variant of the problem where the points within distance $r$ and $cr$ of the query can be either counted or discarded. In particular for an approximation factor $c \geq 1$, any number between $|X \cap B(q, r)|$ and $|X \cap B(q, cr)|$ is valid.

**Differential Privacy (DP):** For two datasets $X$ and $X'$, we use $X \sim X'$ to mean that they are neighboring datasets, i.e., one of them can be obtained from the other by an addition of a single element (in our problem a point).

**Definition 2.1.** *(Differential Privacy [DR+14]). For $\epsilon > 0, \delta \geq 0$, a randomized algorithm $\mathcal{A}$ is $(\epsilon, \delta)$-differentially private ($(\epsilon, \delta)$-DP) if for any two datasets $X \sim X'$, and any possible outcome of the algorithm $S \subseteq Range(\mathcal{A})$, $\mathbb{P}[\mathcal{A}(X) \in S] \leq e^\epsilon \cdot \mathbb{P}[\mathcal{A}(X') \in S] + \delta$. When $\delta = 0$, the algorithm is said to have pure differential privacy.*

The sensitivity of a function $f$ is defined to be $\Delta_f = \max_{X \sim X'} |f(X) - f(X')|$.

We use $\text{Lap}(\lambda)$ to denote the *Laplace* distribution with parameter $\lambda$ with PDF $\mathbb{P}[Z = z] = \frac{1}{2\lambda} e^{-|z|/\lambda}$, which has mean 0 and variance $\lambda^2$. We also use $\text{TLap}(\Delta, \epsilon, \delta)$ to denote the *Truncated Laplace* distribution with PDF proportional to $e^{-|z| \cdot \epsilon / \Delta}$ on the region $[-B, B]$, where $B = \frac{\Delta}{\epsilon} \cdot \log\left(1 + \frac{e^\epsilon - 1}{2\delta}\right)$.

**Lemma 2.2** ((Truncated) Laplace Mechanism [DR+14, GDGK20]). *Given a numeric function $f$ that takes a dataset $X$ as the input, and has sensitivity $\Delta$, the mechanism that outputs $f(X) + Z$ where $Z \sim \text{Lap}(\Delta/\epsilon)$ is $(\epsilon, 0)$-DP. In addition, if $\epsilon, \delta \leq \frac{1}{2}$, $f(X) + Z$, where $Z \sim \text{TLap}(\Delta, \epsilon, \delta)$, is $(\epsilon, \delta)$-DP. Moreover, the Truncated Laplace mechanism is always accurate up to error $B$.*

## 3 Upper Bound

In this section, we describe the algorithm and perform the majority of the analysis for Theorem 1.1. We defer certain parts to Appendix A, and finish the proof there.

### 3.1 Setup

The algorithm we develop is inspired by the data-independent approximate nearest neighbor algorithm in [ALRW17]. We will use similar notation to their paper as well.

Define $S^{d-1}$ to be the $(d-1)$-dimensional unit sphere in $\mathbb{R}^d$. For any parameter $r \in (0, 2)$, let $\alpha(r) = 1 - \frac{r^2}{2}$ and $\beta(r) = \sqrt{1 - \alpha(r)^2}$ be the cosine and the sine, respectively, of the angle between two points in $S^{d-1}$ of distance $r$. For any parameter $\eta > 0$, define

$$F(\eta) := \mathbb{P}_{g \sim \mathcal{N}(0,I)}[\langle g, u \rangle \geq \eta],$$

where $u \in S^{d-1}$ is an arbitrary point on the unit sphere. Note that $F(\eta)$ is independent of $u$ by the spherical symmetry of Gaussians, and in fact equals the probability that a univariate standard Normal exceeds $\eta$. Next, for any parameters $r \in (0, 2)$ and $\eta_q, \eta_u \in \mathbb{R}$, we define

$$G(r, \eta_q, \eta_u) := \mathbb{P}_{g \sim \mathcal{N}(0,I)}[\langle g, q \rangle \geq \eta_q \text{ and } \langle g, p \rangle \geq \eta_u],$$

where $p, q \in S^{d-1}$ are points that have distance exactly $r$ between them. As with $F$, $G$ has no dependence on the specific points $p, q$ but only on the distance $r = \|p - q\|_2$ between them. Note also that $G(r, \eta_q, \eta_u)$ is non-increasing in $r$, even if $\eta_q$ or $\eta_u$ is negative.

Next, we note the following well-known bounds for $F$.

**Algorithm 1** Data Structure
___

1: **Input:** data $X$ of size at most $n$, privacy parameters $\epsilon, \delta$, parameters $T, K, \eta_u$.
2: **Output:** data structure $\mathcal{D} = (\mathcal{T}, \{g_v\}_{v \in \mathcal{T}}, \{c_v\}_{v \text{ leaf} \in \mathcal{T}})$.
3: Create $\mathcal{T}$: $T$-ary tree of depth $K$, with root $v_{\text{root}}$.
4: **for** each node $v \in \mathcal{T}$ **do**
5:     $g_v \leftarrow \mathcal{N}(0, I_d)$. {Random $d$-dimensional Gaussian}
6:     $c_v \leftarrow \emptyset$. {Counts number of points assigned to each node}
7: **for** point $p \in X$ **do**
8:     $v \leftarrow v_{\text{root}}$.
9:     **for** $\ell = 0$ to $K - 1$ **do**
10:        **if** $\text{depth}(v) < \ell$ **then**
11:           **BREAK** {Break out of "for $\ell = 0$ to $K - 1$" loop ($p$ failed to map to a child)}
12:        **for** $i = 1$ to $T$ **do**
13:           $v_i \leftarrow i$th child of $v$
14:           **if** $\langle g_{v_i}, p \rangle \geq \eta_u$ **then**
15:              $c_{v_i} \leftarrow c_{v_i} + 1$
16:              $v \leftarrow v_i$
17:              **BREAK** {Break out of "for $i = 1$ to $T$" loop}
18: **for** each leaf node $v \in \mathcal{T}$ **do**
19:     $c_v \leftarrow c_v + \text{TLap}(1, \frac{\epsilon}{2}, \frac{\delta}{2})$.
20:     **if** $c_v \leq \frac{2}{\epsilon} \cdot \log\left(1 + \frac{e^{\epsilon/2} - 1}{\delta}\right)$ **then**
21:        $c_v \leftarrow 0$
___

**Proposition 3.1.** *For $\eta \geq 1$,*

$$\Omega\left(\frac{1}{\eta}\right) \cdot e^{-\eta^2/2} \leq F(\eta) \leq e^{-\eta^2/2}.$$

*In addition, as $\eta \to \infty$, $F(\eta) \leq o(1) \cdot e^{-\eta^2/2}$.*

### 3.2 Algorithm for data on the sphere

We now present an algorithm that obtains the desired privacy and accuracy guarantees, assuming both the data points and query points are on a unit sphere and the radius $r$ is not too small. The initial algorithm description will not necessarily run in polynomial time, but we will later describe a simple modification to make this algorithm efficient. Finally, in Appendix A.2 we will modify the algorithm to work for general $r$ and in Euclidean space, rather than just on the unit sphere. This follows from known embedding methods [BRS11], which is why we defer it to the Appendix.

**Data Structure:** We describe the data structure in words here, provide pseudocode in Algorithm 1. Fix $K$ and $T$ to be positive integers, to be set later. We generate a data structure, which is a $T$-ary tree of depth $K$. We index the levels as $0, 1, \ldots, K$, where level 0 is the root and level $K$ consists of the $T^K$ leaves. For each node $v$ except the root, we generate an independent $d$-dimensional standard Gaussian $g_v \sim \mathcal{N}(0, I)$. In addition, we will define two parameters $\eta_q$ and $\eta_u$, to be set later.

The tree nodes at level $\ell$ will partition the unit sphere $S^{d-1}$ into $T^\ell$ regions, plus an extra "remainder" region of points that are not mapped into any of the $T^\ell$ regions. At level 0, all points in $S^{d-1}$ are sent to the single region indexed by the root. For any node $v$ in the tree of depth $0 \leq \ell \leq K - 1$ and with children $v_1, \ldots, v_K$, the region $P_v$ is partitioned into $P_{v_1}, \ldots, P_{v_T}$[1] as follows. A point $p \in P_v$ is sent to $P_{v_i}$ if $i \leq T$ is the smallest index such that $\langle g_{v_i}, p \rangle \geq \eta_u$. Note that some points in $P_v$ may not be sent to any of $P_{v_1}, \ldots, P_{v_T}$. For each leaf node $v$, we store an approximate count $c_v = |X \cap P_v| + \text{TLap}(1, \epsilon, \delta)$, where we added Truncated Laplace noise. If the noised count $c_v$ is too small (at most $\frac{\Delta}{\epsilon} \cdot \log\left(1 + \frac{e^\epsilon - 1}{\delta}\right)$), we replace the count $c_v$ with 0.

___

[1] We remark that the partition is technically not a full partition, because some data points may not be sent to any $P_{v_i}$.

---

**Algorithm 2** ANSWER$(\mathcal{D}, \eta_q, q, v)$: Answering a query

---

1: **Input:** data structure $\mathcal{D} = (\mathcal{T}, \{g_v\}_{v \in \mathcal{T}}, \{c_v\}_{v \, \text{leaf} \in \mathcal{T}})$, parameter $\eta_q$, query point $q$, node $v$.
2: **if** $\langle g_v, q \rangle \geq \eta_u$ **then**
3:     **if** $v$ is a leaf node **then**
4:         **Return** $c_v$
5:     **else**
6:         $ans = 0$
7:         **for** $i = 1$ to $T$ **do**
8:             $v_i \leftarrow i$th child of $v$
9:             $ans \leftarrow ans + \text{ANSWER}(\mathcal{D}, \eta_q, q, v_i)$
10:         **Return** $ans$

---

**Answering a query:** We describe the algorithm in words here, provide pseudocode in Algorithm 2. Given a query $q$, we "send" $q$ to every leaf node $v = v_K$ such that the path $v_1, v_2, \ldots, v_K$ from the root $v_{\text{root}}$ satisfies $\langle g_{v_i}, q \rangle \geq \eta_q$ for all $1 \leq i \leq K$. Hence, each query $q$ will correspond to a subset $V_q$ of leaf nodes. The response to the query $q$ will be

$$\sum_{v \in V_q} c_v.$$

To improve the accuracy of the algorithm, we repeat this procedure $O(\log m)$ times if there are $m$ queries, each initialized to $(\epsilon', \delta')$-DP for $\epsilon' = \epsilon / O(\log m)$ and $\delta' = \delta / O(\log m)$. The overall algorithm will still be $(\epsilon, \delta)$-DP, by the well-known Basic Composition theorem [DR$^+$14, Corollary 3.15]. For each query, we output the median response of the individual data structure responses. In the analysis, we will ignore this amplification procedure, as by a Chernoff bound it suffices to show that each individual query is answered accurately with at least $2/3$ probability.

### 3.3 Analysis

**Privacy:** The analysis of privacy is quite simple. First, note that the responses to all of the queries do not depend on the data directly, but only depend on the counts $\{c_v\}$. If we let $\hat{c}_v(X) := |X \cap P_v|$ and $\tilde{c}_v(X) := |X \cap P_v| + \text{TLap}(1, \epsilon, \delta)$, then note that $c_v(X)$ is entirely dependent on $\tilde{c}_v(X)$. Hence, it suffices to show the following.

**Lemma 3.2** (Privacy). *The set* $\{\tilde{c}_v(X)\}_{v \, \text{leaf} \in \mathcal{T}}$ *is* $(\epsilon, \delta)$-*DP with respect to the data* $X$. *Hence, the entire algorithm is also* $(\epsilon, \delta)$-*DP, even if we have an arbitrary number of queries.*

*Proof.* Consider two adjacent datasets $X, X'$, i.e., where we either removed or added a single data point from $X$ to obtain $X'$. Note that the construction partitioning of the dataset into leaves is not dependent on the data, but merely on random vectors $g_v$ over all nodes $v$. Therefore, if we condition on the partitioning of the entire space, the values $\hat{c}_v(X) = |X \cap P_v|$ and $\hat{c}_v(X') = |X' \cap P_v|$ are the same for all but at most one leaf $v$, which changes by at most 1. The sensitivity of each such $\hat{c}_v(X)$ is at most 1, which means that we have $(\epsilon, \delta)$-privacy loss from $\tilde{c}_v(X)$. Because this happens for a single choice of $v$, we have $(\epsilon, \delta)$-privacy loss in total. $\square$

**Accuracy:** We will only consider the accuracy with respect to a fixed query $q$ and a single copy of the data structure, and show accuracy holds in expectation. For a set of $m$ queries, since we repeat this data structure $O(\log m)$ times and use the median estimate, a Chernoff bound implies all queries will be answered accurately with at least $99\%$ probability.

We prove the following lemma to bound accuracy.

**Lemma 3.3** (Accuracy, assuming parameters are set properly). *Suppose the following hold, for some fixed choice of parameters* $c, \rho, K, T, \eta_q, \eta_u$:

1. $(T \cdot F(\eta_q))^K \leq n^{\rho + o(1)}$. *(Additive error due to Laplace noise).*

2. $e^{-TF(\eta_u)} = o(1/K)$, *and* $\frac{G(r, \eta_q, \eta_u)}{F(\eta_u)} = 1 - o(1/K)$. *(Multiplicative error due to not including points within $r$ of $q$).*

3. $\left(\frac{G(cr,\eta_q,\eta_u)}{F(\eta_u)}\right)^K \leq n^{-1+\rho+o(1)}$. *(Additive error due to including points not within $cr$ of $q$).*

*Then, if $ans$ is the response when querying a fixed $q$ on a dataset $X$, with probability at least $2/3$, $ans$ is between*

$$(1-o(1)) \cdot |X \cap B_2(q,r)| - O\left(\frac{1}{\epsilon} \cdot \log\frac{1}{\delta} \cdot n^{\rho+o(1)}\right)$$

*and*

$$(1+o(1)) \cdot |X \cap B_2(q,cr)| + O\left(\frac{1}{\epsilon} \cdot \log\frac{1}{\delta} \cdot n^{\rho+o(1)}\right).$$

*In other words, we solve the $(c,r)$-near neighbor counting problem with probability at least $2/3$, up to a multiplicative $1 \pm o(1)$ factor and an additive $O\left(\frac{1}{\epsilon} \cdot \log\frac{1}{\delta} \cdot n^{\rho+o(1)}\right)$ factor.*

In Subsection 3.4, we will show how to set $T, K, \eta_q, \eta_u$ so that the three conditions above hold.

*Proof.* Our error comes from two sources. The first is the Truncated Laplace noise that we add, which we add in $|V_q|$ locations to compute the answer to query $q$. This adds an additive noise of at most $O(\frac{1}{\epsilon} \cdot \log\frac{1}{\delta} \cdot \mathbb{E}|V_q|)$ in expectation, as $|c_v - |X \cap P_v|| \leq O(\frac{1}{\epsilon} \cdot \log\frac{1}{\delta})$ with probability 1.

To bound $\mathbb{E}|V_q|$, note that it simply equals the sum of the probabilities that $q \in P_v$ for each leaf $v$, which is $T^K$ times the probability that $q \in P_v$ for a fixed $v$. This probability is just $F(\eta_q)^K$ since each Gaussian along the path to $v$ is independent. Hence, the first source error has magnitude at most $O\left(\frac{1}{\epsilon} \cdot \log\frac{1}{\delta} \cdot (T \cdot F(\eta_q))^K\right) \leq O\left(\frac{1}{\epsilon} \cdot \log\frac{1}{\delta} \cdot n^{\rho+o(1)}\right)$ in expectation.

Hence, if we let $\widehat{ans}$ be the response we would have received if we did not add Laplace noise (i.e., if we used $\hat{c}_v$ instead of $c_v$), then $\mathbb{E}[|ans - \widehat{ans}|] \leq O\left(\frac{1}{\epsilon} \cdot \log\frac{1}{\delta} \cdot n^{\rho+o(1)}\right)$, so $|ans - \widehat{ans}| \leq O\left(\frac{1}{\epsilon} \cdot \log\frac{1}{\delta} \cdot n^{\rho+o(1)}\right)$ with probability at least 0.9 by Markov's inequality.

The second source of error is that $\widehat{ans}$ counts the number of points in $X$ that are in $P_v$ for some $v \in V_q$, whereas we actually want to count the number of points in $X$ that are within distance $r$ of $q$ (tolerating the inclusion of points up to distance $cr$ of $q$). Hence, we need to make sure of two things, corresponding to not underestimating or overestimating, respectively:

1. Most points in $X$ within distance $r$ of $q$ are mapped to some $P_v$ where $v \in V_q$.

2. Few points in $X$ that are not within distance $cr$ of $q$ are mapped to some $P_v$ where $v \in V_q$.

This way, we will show that with high probability,

$$(1-o(1)) \cdot |X \cap B_2(q,r)| \leq \widehat{ans} \leq |X \cap B_2(q,cr)| + O\left(n^{\rho+o(1)}\right),$$

which is sufficient.

For any point $p$ of distance $s$ from $q$, we compute the probability that $p \in P_v$ for some $v \in V_q$. Let's start by assuming $K = 1$, i.e., we have a depth-1 tree with leaves labeled $1, 2, \ldots, T$. In this case there are two possibilities for failure: either $p$ is not mapped to any $P_i$ for $1 \leq i \leq T$, or $p$ is mapped to some $P_i$ but $q$ is not sent there. Suppose $p$ is mapped to $P_1$: this means $\langle g_1, p \rangle \geq \eta_u$. Conditioned on this event, the probability that $\langle g_1, q \rangle \geq \eta_q$ is precisely $\mathbb{P}[\langle g_1, q \rangle \geq \eta_q | \langle g_1, p \rangle \geq \eta_u] = \frac{\mathbb{P}[\langle g_1,q\rangle \geq \eta_q, \langle g_1,p\rangle \geq \eta_u]}{\mathbb{P}[\langle g_1,p\rangle \geq \eta_u]} = \frac{G(s,\eta_q,\eta_u)}{F(\eta_u)}$. Suppose $p$ is mapped to $P_i$ for $i \geq 2$. Then, the probability that $\langle g_i, q \rangle \geq \eta_q$ is $\mathbb{P}[\langle g_i, q \rangle \geq \eta_q | \langle g_i, p \rangle \geq \eta_u, \langle g_j, p \rangle < \eta_u \forall j < i]$. But even if $p, q$ are fixed, then the values of $\langle g_j, p \rangle$ for $j < i$ are independent of $(\langle g_i, p \rangle, \langle g_i, q \rangle)$. Hence, we may remove the conditioning on $\langle g_j, p \rangle < \eta_u \forall j < i$, to again say that the conditional probability is $\frac{G(s,\eta_q,\eta_u)}{F(\eta_u)}$. Hence, the probability that $p \in P_v$ for $v \in V_q$, conditioned on $p$ being in some $P_v$, is $\frac{G(s,\eta_q,\eta_u)}{F(\eta_u)}$. The probability that $s \notin P_v$ for any $v$ is $(1 - F(\eta_u))^T$, so the overall probability that $p \in P_v$ for $v \in V_q$, for $K = 1$, is

$$\left(1 - (1 - F(\eta_u))^T\right) \cdot \frac{G(s,\eta_q,\eta_u)}{F(\eta_u)}.$$

Note that $1 - e^{-TF(\eta_u)} \leq 1 - (1 - F(\eta_u))^T \leq 1$.

For general depths, note that the Gaussians selected at each level are independent of the previous levels. Hence, the success probability simply raises to the power of $K$, or equals

$$\left(1 - (1 - F(\eta_u))^T\right)^K \cdot \left(\frac{G(s, \eta_q, \eta_u)}{F(\eta_u)}\right)^K.$$

Note that $1 \geq (1 - e^{-TF(\eta_u)})^K \geq 1 - K \cdot e^{-TF(\eta_u)}$. Hence, the probability that a point $p$ of distance $s$ from $q$ is mapped to some $P_v$ where $v \in V_q$ is between $\left(1 - K \cdot e^{-TF(\eta_u)}\right) \cdot \left(\frac{G(s, \eta_q, \eta_u)}{F(\eta_u)}\right)^K$ and $\left(\frac{G(s, \eta_q, \eta_u)}{F(\eta_u)}\right)^K$.

Because $G(s, \eta_q, \eta_u)$ is an increasing function in $s$, the probability at any point of distance $s \leq r$ from $q$ is not included in the count $\widehat{ans}$ is at least $\left(1 - K \cdot e^{-TF(\eta_u)}\right) \cdot \left(\frac{G(r, \eta_q, \eta_u)}{F(\eta_u)}\right)^K = 1 - o(1)$, as we are assuming that $e^{-T \cdot F(\eta_u)} = o(1/K)$ and $\frac{G(r, \eta_q, \eta_u)}{F(\eta_u)} = 1 - o(1/K)$. Thus, by Markov's inequality, with probability at least 0.9, $\widehat{ans} \geq (1 - o(1)) \cdot |X \cap B_2(q, r)|$. Next, the probability that any point of distance $s \geq cr$ from $q$ is included in the count $\widehat{ans}$ is at most $\left(\frac{G(cr, \eta_q, \eta_u)}{F(\eta_u)}\right)^K$, which we assume is at most $n^{-1+\rho+o(1)}$. Because there are at most $n$ points, by Markov's inequality, $\widehat{ans} \leq |X \cap B_2(q, cr)| + n^{\rho+o(1)}$ with probability at least 0.9. $\qquad\square$

**Runtime:** While we have written the algorithm to create a $T^K$-sized tree, we can speed up the implementation by only including the necessary parts of the data structure.

We can bound the runtime as follows. Naively, for preprocessing, it takes space and time $O(T^K \cdot d)$ to construct the tree and generate and store the Gaussians. Next, for each point $p$, if $p \in P_v$, it takes up to $O(T \cdot d)$ time to determine which child of $v$ is that $p$ will be sent to. Since $p$ is sent to only one node in each partition, this takes total time $O(K \cdot T \cdot d)$ per point, which means $O(n \cdot K \cdot T \cdot d)$ total time to partition the points in $X$. To improve the preprocessing time, we do not generate the full tree: rather, we only generate a node of the tree (with a corresponding Gaussian $g_v$) if we need to check whether some data point $p \in X$ is sent to the node, which means we can improve the preprocessing time to $O(n \cdot K \cdot T \cdot d)$, without the extra term of $O(T^K \cdot d)$. Note that if some leaf node $v$ was not created in our modified implementation, then no point would have been sent to some partition piece $P_v$, which means $|X \cap P_v| + \text{TLap}(1, \frac{\epsilon}{2}, \frac{\delta}{2}) \leq \frac{2}{\epsilon} \cdot \log\left(1 + \frac{e^{\epsilon/2}-1}{\delta}\right)$. So, we would have set $c_v$ to be 0 anyway, which means that this modification does not affect any of the responses to queries.

To answer a query $q$, if $q \in P_v$ (where $v$ is possibly not a leaf node) it takes $O(T \cdot d)$ time to determine all of the children of $v$ that $q$ will be sent to. In expectation, $q$ is sent to $(TF(\eta_q))^\ell$ nodes at level $\ell$, which means at level $\ell + 1$ we need to check up to $(TF(\eta_q))^\ell \cdot T$ nodes in expectation. (If a node $w$ is not created in our implementation, we do not have to check it, since we know $c_v = 0$ for any leaf $v$ that is a descendant of $w$.) Therefore, the total time it takes to determine all leaf nodes $q$ is sent to, in expectation, is $O\left(d \cdot \sum_{\ell=0}^{K-1} (TF(\eta_q))^\ell \cdot T\right) = O((TF(\eta_q))^K \cdot K \cdot T \cdot d)$. Finally, we can add $c_v$ over all leaf nodes $v$ that $q$ is sent to, which takes an additional $O((TF(\eta_q))^K)$ time in expectation.

Hence, we have the following lemma.

**Lemma 3.4** (Runtime). *The total preprocessing time of the data structure is $O(n \cdot K \cdot T \cdot d)$, and the expected time needed to answer each query is $O((TF(\eta_q))^K \cdot K \cdot T \cdot d)$.*

### 3.4 Parameter Settings and Finishing the Accuracy/Runtime Analysis

In this section, we set parameters properly so that Lemmas 3.3 and 3.4 match the goals of Theorem 1.1. We recall that $c > 1$ is a fixed constant representing the approximation ratio. We assume that $d = (\log n)^{O(1)}$ and $r = \Theta\left(\frac{1}{(\log n)^{1/8}}\right)$. For fixed $c$ and $r$ as set above, we set the parameters as follows:

- $K := \sqrt{\ln n}$.
- $\eta_u := \sqrt{\frac{\ln n}{K}} \cdot \frac{\lambda}{r}$, for some constant $\lambda = \lambda(c)$, that we will set later.

- $\eta_q := \alpha(r) \cdot \eta_u - 2\beta(r) \cdot \sqrt{\ln K}$. Recall that $\alpha(r) = 1 - \frac{r^2}{2}$ and $\beta(r) = \sqrt{1 - \alpha(r)^2}$.
- $T := 10 \ln K / F(\eta_u)$.

First, we must show that our parameter choices imply that the algorithm is accurate. We prove the following lemma, for which the proof is deferred to Appendix A.

**Lemma 3.5** (Accuracy, completed). *For an appropriate choice of $\lambda(c) = \frac{2\sqrt{2}c}{c^2+1}$, all three conditions in Lemma 3.3 hold, with $\rho = \frac{4c^2}{(c^2+1)^2} = O(\frac{1}{c^2})$. Hence, Lemma 3.3 holds without the conditions, for $\rho = \frac{4c^2}{(c^2+1)^2}$.*

Finally, we show that the runtime is good, under our parameter choices.

**Lemma 3.6** (Runtime, completed). *For the parameters we have defined, assuming $r = \Theta\left(\frac{1}{(\log n)^{1/8}}\right)$, the total preprocessing time is $n^{1+o(1)} \cdot d$ and the expected time to answer each query is $n^{\rho+o(1)} \cdot d$, for $\rho = \frac{4c^2}{(c^2+1)^2}$.*

*Proof.* The preprocessing time is $O(n \cdot K \cdot T \cdot d)$. We know that $T = \frac{10 \ln K}{F(\eta_u)}$, and that $\frac{1}{F(\eta_u)} \leq \Omega(\eta_u) \cdot e^{\eta_u^2/2} \leq (\log n)^{O(1)} \cdot e^{\ln n \cdot \lambda^2 / (r^2 \cdot K)}$. By our settings of $r = \Theta((\log n)^{-1/8})$ and $K = \sqrt{\ln n}$, and since $\lambda$ is a constant, this equals $n^{\Theta(1/(\log n)^{1/4})} = n^{o(1)}$. Hence, $1/F(\eta_u) = n^{o(1)}$, and since $K = O(\sqrt{\log n})$, the total preprocessing time is $n^{1+o(1)} \cdot d$.

The time to answer each query is $O((TF(\eta_q))^K \cdot K \cdot T \cdot d)$. By Lemma 3.5, we know that $(TF(\eta_q))^K = n^{\rho+o(1)}$. Moreover, we know that $K = (\log n)^{O(1)}$, and we already saw that $T = 10 \ln K / F(\eta_u) = n^{o(1)}$. Hence, the time to answer each query is $n^{\rho+o(1)} \cdot d$. $\qquad\square$

Lemmas 3.2, 3.5, and 3.6 will end up being sufficient to prove Theorem 1.1 if we assume the data lies on a $d = (\log n)^{O(1)}$-dimensional unit sphere, and if $r = \Theta((\log n)^{1/8})$. To finish the analysis, we must show that this assumption can be made without loss of generality, which will follow from an embedding argument of [BRS11]. We defer this argument, and the proof of the remainder of Theorem 1.1, to Appendix A.2.

*Remark.* We note that the data structure can be implemented so that extra space is only $O(nd)$. Instead of generating and storing the Gaussians $g_v$ independently, we note that in each level $\ell$, each group of $T$ siblings can reuse exactly the same random set of $T$ Gaussians. The analysis goes through mutatis mutandis as independence between different branches is not needed.

**Acknowledgements:** PI was supported by the NSF TRIPODS program (award DMS-2022448), Simons Investigator Award, and GIST-MIT Research Collaboration grant.

SN is supported by a NSF Graduate Fellowship (Grant No. 1745302) and a Google PhD Fellowship.

AA supported in part by NSF (CCF2008733) and ONR (N00014-22-1-2713).

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

# A  Omitted Proofs from Section 3

## A.1  Completion of Accuracy Analysis

Here, we complete the accuracy analysis by proving Theorem 3.5.

*Proof of Theorem 3.5.* To bound $(T \cdot F(\eta_q))^K$, note that $(10 \ln K)^K \le n^{o(1)}$ by our setting of $K = \sqrt{\ln n}$. Therefore, it suffices to show that $(F(\eta_q)/F(\eta_u))^K \le n^{\rho + o(1)}$. However, by Proposition 3.1, we can upper bound $F(\eta_q)/F(\eta_u) \le O(\eta_u) \cdot e^{(\eta_u^2 - \eta_q^2)/2}$. But we know that $\eta_u \le (\log n)^{O(1)}$, which implies that $O(\eta_u)^K \le n^{o(1)}$. Hence, it suffices to bound $(e^{(\eta_u^2 - \eta_q^2)/2})^K$. Indeed, we can write

$$
\begin{aligned}
\eta_u^2 - \eta_q^2 &= \eta_u^2 - (\alpha(r) \cdot \eta_u - 2\beta(r) \cdot \sqrt{\ln K})^2 \\
&\le (1 - \alpha(r)^2) \cdot \eta_u^2 + 4\alpha(r)\beta(r)\sqrt{\ln K} \cdot \eta_u \\
&\le \lambda^2 \cdot \frac{\ln n}{K} + 4\lambda \cdot \sqrt{\frac{\ln n \cdot \ln K}{K}} \\
&\le \left( \lambda^2 \cdot \frac{\ln n}{K} \right) \cdot (1 + o(1)).
\end{aligned}
$$

Above, the third line follows from the fact that $1 - \alpha(r)^2 \le r^2$ and $\alpha(r) \cdot \beta(r) \le \beta(r) \le r$, and the fourth line follows from our definition of $K$. Therefore,

$$
\begin{aligned}
(T \cdot F(\eta_q))^K &\le n^{o(1)} \cdot \exp\left( \frac{1}{2} \cdot \left( \lambda^2 \cdot \frac{\ln n}{K} \right) \cdot (1 + o(1)) \right)^K \\
&= n^{o(1)} \cdot \exp\left( \frac{\lambda^2}{2} \cdot \ln n \cdot (1 + o(1)) \right) \\
&= n^{\lambda^2/2 + o(1)}.
\end{aligned}
$$

Next, note that $e^{-TF(\eta_u)} = e^{-10 \log K} = K^{-10} = o(1/K)$, since $K = \omega(1)$.

Next, we bound $\frac{G(r, \eta_q, \eta_u)}{F(\eta_u)}$. Since $G(r, \eta_q, \eta_u) = \mathbb{P}[\langle g, q \rangle \ge \eta_q \text{ and } \langle g, p \rangle \ge \eta_u]$ for $\|p - q\|_2 = r$, we can rewrite this as $\frac{G(r, \eta_q, \eta_u)}{F(\eta_u)} = \mathbb{P}[\langle g, q \rangle \ge \eta_q | \langle g, p \rangle \ge \eta_u]$ for $\|p - q\|_2 = r$. Since $p, q$ are unit vectors, we can write $q = \alpha(r) \cdot p + \beta(r) \cdot p'$, where $p'$ is a unit vector orthogonal to $p$. Hence, if we let $Y = \langle g, p \rangle$ and $Z = \langle g, p' \rangle$, then $Y, Z$ are independent standard Gaussians and our goal is to bound $\mathbb{P}[\alpha(r) \cdot Y + \beta(r) \cdot Z \ge \eta_q | Y \ge \eta_u]$. But, $\eta_q = \alpha(r) \cdot \eta_u - 2\beta(r) \cdot \sqrt{\ln K}$, which means if $Z \ge -2\sqrt{\ln K}$, then $Y \ge \eta_u$ automatically implies $\alpha(r) \cdot Y + \beta(r) \cdot Z \ge \eta_q$. Therefore, our probability is at least $\mathbb{P}(Z \ge -2\sqrt{\ln K}) = 1 - \mathbb{P}(Z \ge 2\sqrt{\ln K}) = 1 - o(1/K)$, since $\mathbb{P}(Z \ge 2\sqrt{\ln K}) = O(1/K^2)$.

Finally, we must bound $\left( \frac{G(cr, \eta_q, \eta_u)}{F(\eta_u)} \right)^K$. Note that $\frac{G(cr, \eta_q, \eta_u)}{F(\eta_u)} = \mathbb{P}[\langle g, q \rangle \ge \eta_q | \langle g, p \rangle \ge \eta_u]$ for $\|p - q\|_2 = cr$, which means $q = \alpha(cr) \cdot p + \beta(cr) \cdot p'$. Again, letting $Y = \langle g, p \rangle$ and $Z = \langle g, p' \rangle$, we wish to bound

$$
\frac{G(cr, \eta_q, \eta_u)}{F(\eta_u)} = \mathbb{P}\left[ \alpha(cr) \cdot Y + \beta(cr) \cdot Z \ge \alpha(r) \cdot \eta_u - 2\beta(r) \cdot \sqrt{\ln K} \,\middle|\, Y \ge \eta_u \right]. \tag{1}
$$

We can rewrite this as

$$
\mathbb{P}\left[ Z \ge \frac{(\alpha(r) - \alpha(cr)) \cdot \eta_u - 2\beta(r) \cdot \sqrt{\ln K}}{\beta(cr)} - \frac{\alpha(cr) \cdot (Y - \eta_u)}{\beta(cr)} \,\middle|\, Y \ge \eta_u \right] \tag{2}
$$

If we write $W = Y - \eta_u$, note that the PDF of $W$ is $\frac{1}{\sqrt{2\pi}} \cdot e^{-(W + \eta_u)^2/2}$. The probability that $W \ge 0$ is between $\Omega\left( \frac{1}{\eta_u} \right) \cdot e^{-\eta_u^2/2}$ and $e^{-\eta_u^2/2}$, which means

$$
\mathbb{P}(W \ge w | W \ge 0) \le \frac{e^{-(\eta_u + w)^2/2}}{\Omega(1/\eta_u) \cdot e^{-\eta_u^2/2}} \le O(\eta_u) \cdot e^{-w^2/2 - w \cdot \eta_u} \le O(\eta_u) \cdot e^{-\eta_u \cdot w}.
$$

Next, a simple calculation tells us that

$$\frac{\alpha(cr)}{\beta(cr)} = \frac{1 - \frac{(cr)^2}{2}}{cr \cdot \sqrt{1 - \frac{(cr)^2}{4}}} = \frac{1}{cr} \cdot (1 \pm o(1)),$$

and

$$\frac{(\alpha(r) - \alpha(cr)) \cdot \eta_u - 2\beta(r) \cdot \sqrt{\ln K}}{\beta(cr)} = \frac{\left(\frac{c^2-1}{2}\right) r^2 \cdot \sqrt{\frac{\ln n}{K}} \cdot \frac{\lambda}{r} - 2r\sqrt{\ln K}(1 \pm o(1))}{cr(1 \pm o(1))}$$

$$= \frac{\sqrt{\frac{\ln n}{K}} \cdot r \cdot \frac{(c^2-1)\lambda}{2} \cdot (1 \pm o(1))}{cr(1 \pm o(1))}$$

$$= \sqrt{\frac{\ln n}{K}} \cdot \frac{(c^2 - 1)\lambda}{2c} \cdot (1 \pm o(1)).$$

Let $r' = \frac{\beta(cr)}{c \cdot \alpha(cr)} = r(1 \pm o(1))$. Then, we can therefore upper bound (2) as at most

$$\mathbb{P}\left[Z + \frac{W}{cr'} \geq \sqrt{\frac{\ln n}{K}} \cdot \frac{(c^2 - 1)\lambda}{2c} \cdot (1 - o(1)) \,\Big|\, W \geq 0\right]. \tag{3}$$

The CDF of $\frac{W}{cr'}$ conditioned on $W \geq 0$ is bounded above by $O(cr' \cdot \eta_u) \cdot e^{-cr' \cdot \eta_u \cdot w} \leq (\log n)^{O(1)} \cdot e^{-S \cdot w}$, where $S = c \cdot \frac{r'}{r} \cdot \lambda \cdot \sqrt{\ln n / K}$. In addition, we write $p_Z$ to represent the PDF of $Z$ (which is a standard Gaussian) and $T = \sqrt{\ln n / K} \cdot (c^2 - 1)\lambda/(2c) \cdot (1 - o(1))$. Note that $S > T$, since it suffices for $c > \frac{c^2-1}{2c} \cdot (1 + o(1))$, which is true for any constant $c$. Then, we have that (3) is at most

$$\int_{-\infty}^{\infty} \mathbb{P}\left(\frac{W}{cr'} \geq w \,\Big|\, W \geq 0\right) \cdot p_Z(T - w)dw \leq (\log n)^{O(1)} \cdot \int_0^{\infty} e^{-S \cdot w} \cdot e^{-(T-w)^2/2}dw + \int_{-\infty}^0 e^{-(T-w)^2/2}dw$$

$$\leq (\log n)^{O(1)} \cdot \int_0^{\infty} e^{-T^2/2 - (S-T)w - w^2/2}dw + \int_{-\infty}^0 e^{-T^2/2 + Tw - w^2/2}dw$$

$$\leq (\log n)^{O(1)} \cdot e^{-T^2/2} \cdot \int_{-\infty}^{\infty} e^{-w^2/2}$$

$$= (\log n)^{O(1)} \cdot e^{-T^2/2}.$$

The third line uses the fact that $(S - T)w \geq 0$ when $w \geq 0$ (since $S \geq T$) and that $Tw \leq 0$ when $w \leq 0$ (since $T$ is positive as we assume $c \geq 1$). Therefore,

$$\left(\frac{G(cr, \eta_q, \eta_u)}{F(\eta_u)}\right)^K \leq \left((\log n)^{O(1)} \cdot e^{-(T^2/2)}\right)^K \leq n^{-(c^2-1)^2\lambda^2/(8c^2)+o(1)}.$$

To finish, we will want to set $\frac{\lambda^2}{2} = \rho$ and $\frac{(c^2-1)^2\lambda^2}{8c^2} = 1 - \rho$. Solving gives us $\lambda = \frac{2\sqrt{2}c}{c^2+1}$ and $\rho = \frac{4c^2}{(c^2+1)^2}$. $\qquad \square$

## A.2 Embedding into the Euclidean sphere

In the previous sections, we assumed that the data lied on a reasonably low-dimensional Euclidean sphere, and that $r$ was some reasonably small (but not too small) parameter. In this section, we explain why this assumption can be made without loss of generality, by using a previous work of [BRS11] which allows us to embed the data into a Euclidean sphere.

**Lemma A.1.** *[BRS11, Lemma 6, rephrased] Fix integers $d, d' \geq 1$ and a parameter $0 < \gamma < \frac{1}{2}$. There exists a randomized map $\Theta : \mathbb{R}^d \to S^{d'-1}$, where $S^{d'-1}$ represents the unit sphere in $\mathbb{R}^{d'}$, that can be computed in time $O(d \cdot d')$ for any point $x \in \mathbb{R}^d$, with the following properties for any $x, y \in \mathbb{R}^d$.*

*1. $\|\Theta(x) - \Theta(y)\|_2^2 \leq (1 + \gamma) \cdot \|x - y\|_2^2$ with failure probability at most $\exp\left(\frac{D\gamma^2}{6}\right)$.[2]*

---

[2]The original claim also takes some parameters $\sigma_1, \ldots, \sigma_D$: we will set $D = d'/2$ and every $\sigma_i = 1$. It is clear that the final function $\Theta$ defined in [BRS11] has norm 1 in this case.

2. If $\|x - y\|_2 \leq \sqrt{\gamma}$, then $\|\Theta(x) - \Theta(y)\|_2^2 \geq (1 - \gamma) \cdot \|x - y\|_2^2$ with failure probability at most $\exp\left(-\frac{3D\gamma^2}{128}\right)$.

3. If $\|x - y\|_2 \geq \sqrt{\gamma}$, then $\|\Theta(x) - \Theta(y)\|_2^2 \geq \frac{\gamma}{2}$ with failure probability at most $\exp\left(-\frac{D \cdot \gamma}{128}\right)$.[3]

We will set $d' = O\left(\frac{\log n}{\gamma^2}\right)$, where we recall that $n$ equals the number of data points and $m$ equals the number of queries. Then, if we generate the randomized map $\Theta$ ahead of time, with high probability the conditions in Lemma A.1 hold for all $n$ data points and any fixed query point. (In the final algorithm, we repeat the embedding $O(\log m)$ times to amplify the success probability).

**Corollary A.2.** *Suppose that $(cr)^2 \leq \frac{\gamma}{2}$, and $d' = C \cdot \frac{\log n}{\gamma^2}$ for some sufficiently large constant $C$. Then, with at least $0.99$ probability over the randomized map $\Theta : \mathbb{R}^d \to S^{d'-1}$, for all $x \in X$ and any fixed query $y$, $\|x - y\|_2 \leq r$ implies $\|\Theta(x) - \Theta(y)\|_2 \leq (1 + \gamma) \cdot r$ and $\|x - y\|_2 \geq cr$ implies $\|\Theta(x) - \Theta(y)\|_2 \geq (1 - \gamma) \cdot cr$.*

*Proof.* We assume that $\Theta$ was chosen so that for every $x \in X$ and for the fixed $y$, the three claims in Lemma A.1 all hold.

Fix some pair $(x, y)$, where $x \in X, y \in Q$. First, if $\|x - y\|_2 \leq r$, then since $r \leq \sqrt{\gamma}$, $\|\Theta(x) - \Theta(y)\|_2 \leq (1 + \gamma) \cdot \|x - y\|_2 = (1 + \gamma) \cdot r$. Next, if $c \cdot r \leq \|x - y\|_2 \leq \sqrt{\gamma}$, then $\|\Theta(x) - \Theta(y)\|_2 \geq (1 - \gamma) \cdot \|x - y\|_2 \geq (1 - \gamma) \cdot cr$. Otherwise, if $\|x - y\|_2 \geq \sqrt{\gamma}$, then $\|\Theta(x) - \Theta(y)\|_2 \geq \sqrt{\frac{\gamma}{2}} \geq c \cdot r$. $\square$

We can now finish the proof of Theorem 1.1.

*Proof.* We will prove that we can solve the $(c \cdot \frac{1+\gamma}{1-\gamma}, \frac{r}{1+\gamma})$-near neighbor counting with differential privacy, where $r = (\log n)^{-1/8}$ and $\gamma = (\log n)^{-1/8}$, over Euclidean space. By scaling, this automatically implies the same result for any choice of $r$, and $c$ replaced with $\frac{c(1+\gamma)}{1-\gamma} = c(1 + o(1))$.

We will run $O(\log m)$ independent copies of the data structure and algorithm, where each individual data structure satisfies $(\epsilon', \delta') = \left(\frac{\epsilon}{O(\log m)}, \frac{\delta}{O(\log m)}\right)$-DP. By basic composition, this implies $(\epsilon, \delta)$-DP overall. For each query, we output the median of the responses of each data structure, which by a Chernoff bound will significantly decrease the failure probability.

For each copy of the data structure, we first preprocess the data $X \in \mathbb{R}^d$ by using the random map $\Theta$ from Lemma A.1 to map each point $x \in X$ to $\Theta(x)$, which takes $O(nd \cdot d')$ time. Next, we create the data structure on $\Theta(X)$, which is on a $d' = O\left(\frac{\log n}{\gamma^2}\right) = (\log n)^{O(1)}$-dimensional sphere. To answer any query $q \in \mathbb{R}^d$, we compute $\Theta(q)$ in $O(d \cdot d')$ time, and answer the query $\Theta(q)$ using the data structure created on $\Theta(X)$. By Corollary A.2, with $0.99$ probability, any accurate answer for $(c, r)$-near neighbor counting on $S^{d'-1}$ for a fixed query $\Theta(q)$ is an accurate answer for $\left(\frac{(cr)/(1-\gamma)}{r/(1+\gamma)}, \frac{r}{1+\gamma}\right) = (c \cdot \frac{1+\gamma}{1-\gamma}, \frac{r}{1+\gamma})$-near neighbor counting problem for $q$ over $\mathbb{R}^d$. Moreover, the embedding $\Theta$ is oblivious to the data, which means privacy is preserved.

Now, we have embedded the data and queries onto the unit sphere $S^{d'-1}$ for $d' = (\log n)^{O(1)}$, in $O(nd \cdot d') = \tilde{O}(nd)$ preprocessing time. We may now create the data structure (Algorithm 1) and answer each query (Algorithm 2) on $S^{d'-1}$. We recall that, to improve the failure probability so that every query is answered with high probability, we replaced $(\epsilon, \delta)$ with $(\epsilon', \delta') = (\frac{\epsilon}{O(\log m)}, \frac{\delta}{O(\log m)})$, and we will take the median over $O(\log m)$ independent copies of the data structure, for each query. By Lemmas 3.2 and 3.6, the privacy and runtime conditions of Theorem 1.1 are all satisfied. Moreover, by Lemma 3.5, with probability at least $2/3 - 0.01 \geq 0.6$, for each query $q$, each copy of the data structure is accurate, replacing the additive error $\frac{1}{\epsilon} \log \frac{1}{\delta} \cdot n^{\rho+o(1)}$ with $\frac{\log m}{\epsilon} \log \frac{\log m}{\delta} \cdot n^{\rho+o(1)}$. By a Chernoff bound, the overall algorithm (taking the median of $O(\log m)$ copies) is accurate on each

---

[3] In the original paper [BRS11] they only prove this result for $\|x - y\|_2 \geq \frac{1}{\sqrt{2}}$, the same proof can extend to $\|x - y\|_2 \geq \gamma$, at the cost of reducing the failure probability from exponential in $-D$ to exponential in $-D \cdot \gamma$.

query with probability at least $1 - \frac{1}{100m}$, so it is accurate on all queries simultaneously with at least $0.99$ probability. This concludes the proof. $\qquad\square$

# B   Lower Bound

In this section, we prove Theorem 1.2.

Given $n$ points $z_1, \ldots, z_n \in \{0,1\}^d$, and a series of $m$ queries $B_1, \ldots, B_m$ represented by axis-parallel boxes, we create the matrix $A_{i,j}$ to be the indicator of $B_i$ containing $z_j$. We define the red-blue discrepancy to be $\max_{x \in \{-1,1\}^n} \|Ax\|_\infty$.

First, we have the following result due to Chazelle and Lvov [CL01].

**Theorem B.1.** *[CL01] For sufficiently large $n$ and $d = C \log n$ for a large constant $C$, there exists a set of $n$ points $z_1, \ldots, z_n \in \{0,1\}^d$ and $n$ boxes $B_1, \ldots, B_n$ such that the matrix $\{A_{i,j}\} \in \mathbb{R}^{n \times n}$ has red-blue discrepancy $n^{\Omega(1)} = 2^{\Omega(d)}$.*

Now, we switch to the result of Muthukrishnan and Nikolov [MN12]. They have a slightly more general definition of discrepancy, of $\mathrm{disc}_{p,\alpha}(A) = \min_{x \in \{-1,0,1\}^d, \|x\|_1 \geq \alpha \cdot d} \|Ax\|_p$, so the above result is written as a lower bound on $\mathrm{disc}(A) := \mathrm{disc}_{\infty,1}(A)$. Now, we describe their theorem.

**Theorem B.2.** *[MN12] For all $x \in \{0,1\}^n$, there exist constants $\epsilon, \delta > 0$ such that no mechanism $\mathcal{M} = \{\mathcal{M}_n\}$ that satisfies*

$$\forall x \in \{0,1\}^n, \mathbb{P}\left(\|\mathcal{M}_n(x) - Ax\|_\infty < \frac{\mathrm{disc}(A)}{2}\right) \geq 2/3$$

*is $(\epsilon, \delta)$-DP, where two data points $x, x' \in \{0,1\}^n$ are adjacent if they differ exactly in one coordinate location.*

Given these lemmas, we are ready to prove our main lower bound, Theorem 1.2.

*Proof of Theorem 1.2.* We set $m = n$ and $d = C \log n$ for some constant $C$. By using Theorem B.1, we can construct a public set $\{z_1, \ldots, z_n\}$ of distinct points in $\{0,1\}^d$ and a fixed set of boxes $B_1, \ldots, B_n$, that generate a matrix $A$ with discrepancy $\mathrm{disc}(A) \geq n^\rho$, where $\rho > 0$ is a fixed constant. For any $x \in \{0,1\}^n$, we define the dataset $\mathcal{D}_x$ will be the subset of the public set where $z_i \in \mathcal{D}_x$ if and only if $x_i = 1$. Then, $Ax$ would be an $m = n$ dimensional vector that precisely tells us the number of points in $\mathcal{D}_x$ are in each box. In addition, if $x, x'$ are adjacent in $\{0,1\}^n$, then $D_x, D_{x'}$ are adjacent datasets. Next for any axis-parallel box $B_i$, we create a query point $q_i \in \{0,1\}^d$ as follows. In the $k$th coordinate dimension, if $B_i$ intersects either both or neither of the planes $x_k = 0$ and $x_k = 1$, we set $q_{i,k} = 1/2$. Otherwise, if $B_i$ only intersects the plane $x_k = 0$, we set $q_{i,k} = -1/2$. Finally, if $B_i$ only intersects the plane $x_k = 1$, we set $q_{i,k} = 3/2$.

Suppose that $\mathcal{A}$ is an $(\epsilon, \delta)$-DP algorithm acting on subsets $S$ of $\{0,1\}^d$ of size at most $n$, that returns an estimate to $|S \cap B_\infty(q_i, 1)|$ for all queries $q_i$. Since every $q_i$ has half-integral coordinates, and every data point has integral coordinates, we have that $B_\infty(q_i, 1) = B_\infty(q_i, 1/2) = B_\infty(q_i, 3/2 - \alpha)$ for any $\alpha > 0$, and these all equal $B_i$. So, for any subset $S \subset \{0,1\}^d$,

$$|S \cap B_\infty(q_i, 1)| = |S \cap B_\infty(q_i, 1/2)| = |S \cap B_\infty(q_i, 3/2 - \alpha)| = |S \cap B_i|.$$

Suppose that for any $S \subset \{0,1\}^d$ of size at most $n$, with probability at least $2/3$, the output of $\mathcal{A}(S)$ for each query $q_i$ differs by at most $T$ from $|S \cap B_i|$. Then, we can consider the algorithm $\mathcal{M}$ that acts on an element in $\{0,1\}^n$, where $\mathcal{M}(x) = \mathcal{A}(D_x)$. Since $\mathcal{A}$ is $(\epsilon, \delta)$-DP, this implies that $\mathcal{M}$ is also $(\epsilon, \delta)$-DP, with respect to adjacent elements in $\{0,1\}^n$. Moreover, $(A \cdot x)_i = \sum_j A_{i,j} x_j = |\{j : A_{i,j} = x_j = 1\}| = |D_x \cap B_i|$. Hence, by Theorem B.2, there must exist $x \in \{0,1\}^n$ such that

$$\mathbb{P}\left(\|\mathcal{A}(D_x) - |D_x \cap B_i|\|_\infty < \frac{\mathrm{disc}(A)}{2}\right) = \mathbb{P}\left(\|\mathcal{M}(x) - Ax\|_\infty < \frac{\mathrm{disc}(A)}{2}\right) < \frac{2}{3},$$

so $T \geq \frac{\mathrm{disc}(A)}{2}$. This completes the proof as $\mathrm{disc}(A) \geq n^\rho$. $\qquad\square$

## C  Extensions

$\ell_1$ **balls.**    It is easy to extend our algorithms to ranges that are balls in the $\ell_1$ norm. This relies on the well-known fact that the $\ell_1$ norm embeds (almost) isometrically into a negative-type metric, i.e., into a squared Euclidean metric $(\ell_2)^2$. We use the following constructive version of this fact, from [LN14, Theorem 116] based on [MN04].

**Theorem C.1** ([LN14][4]). *Let $1 \le p < 2$. Let $X \subset \mathbb{R}^d$ be a point set with $\ell_p$-aspect ratio $\Phi$. There is a mapping $f : X \to \mathbb{R}^{d'}$ for $d' = d \cdot \mathrm{poly}(\log \Phi, \log d, 1/\epsilon)$ such that for every $x, y \in X$,*

$$\|x - y\|_p^p \le \|f(x) - f(y)\|_2^2 \le (1 + \epsilon) \|x - y\|_p^p .$$

The above theorem, applied for $p = 1$, reduces approximate range queries for $\ell_1$ balls with approximation factor $c$ to approximate range queries for $(\ell_2)^2$ balls with approximation factor $c/(1 + \epsilon)$ and $d$ replaced by $d'$. The latter problem is equivalent to approximate range queries for $\ell_2$ balls with approximation factor $\sqrt{c/(1 + \epsilon)}$. The final bound follows by following this sequence of reductions.

**Pure Differential Privacy.**    The only reason that our algorithm requires approximate differential privacy is that we use truncated Laplace noise in line 19 of Algorithm 1. We use truncated Laplace noise so that if any leaf node is empty, then the noisy count will be 0 with probability 1. To make the algorithm pure-DP, we modify line 19 to add $\mathrm{Lap}(2/\epsilon)$ noise to the count $c_v$, and in lines 20-21, we replace $c_v$ with 0 whenever the noised count is at most $\frac{10}{\epsilon} \cdot K \log T$. For the parameters in subsection 3.4, this is at most $\frac{1}{\epsilon} \cdot (\log n)^{O(1)}$.

The difficulty, however, is that we need to speed up the runtime from $T^K$, as done in the runtime analysis in subsection 3.3. Specifically, we need to ensure that few nodes are created (i.e., have $c_v = 0$): we prove that, in expectation, this holds. In addition, we need to determine which nodes will have $c_v \ne 0$ efficiently. If so, we only need to create these nodes in the preprocessing, which completes the proof.

First, we determine which nodes every data point $p \in X$ is sent to, in the same fashion as in runtime analysis in subsection 3.3. Suppose there are $n' \le n$ nonempty regions: then there are $N := T^K - n'$ empty regions. For each node $v$, we wish to sample $c_v \sim \mathrm{Lap}(2/\epsilon)$ as the noise, and if $c_v > \frac{10}{\epsilon} \cdot K \log T$, we keep the node $v$ and store $c_v$. Otherwise, we reset $c_v$ back to 0, and do not need to store $v$.

We note that the probability of any fixed $c_v > \frac{4}{\epsilon} \cdot K \log T$ is $\frac{1}{2} \cdot e^{-(\epsilon/2) \cdot (10/\epsilon) \cdot K \log T} = e^{-5K \log T}/2 = T^{-5K}/2$. Let $\alpha := T^{-5K}/2$. We first consider the problem of sampling from $\mathrm{Bin}(N, \alpha)$, which represents the number of empty regions that will have $c_v > 0$. Note that $\mathbb{P}(\mathrm{Bin}(N, \alpha) > 0) = 1 - (1 - T^{-5K}/2)^{T^K - n'} \le T^{-4K}$. By using a repeated squaring approach, we can compute this probability exactly in $\mathrm{poly} \log(T^K) = \mathrm{poly} \log n$ time. Then, we can flip a coin with this probability, and with at most $T^{-4K}$ failure probability, the Binomial equals 0. Otherwise, need to compute the distribution $\mathrm{Bin}(N, \alpha)$ conditioned on it being at least 1. However, we can compute each term $\binom{N}{r} \alpha^r (1 - \alpha)^{N-r} = \mathbb{P}(\mathrm{Bin}(N, \alpha) = r)$ in time at most $O(N^2)$, which thus means we can compute every $\mathbb{P}(\mathrm{Bin}(N, \alpha) = r)$ in time $O(N^3)$. Hence, in the $T^{-4K} \le N^{-4}$ probability event that $\mathrm{Bin}(N, \alpha) > 0$, we can still draw from the conditional distribution in time $O(N^3)$. Hence, in expected time $\mathrm{poly} \log n$, we can draw from $\mathrm{Bin}(N, \alpha)$.

In the case that $\mathrm{Bin}(N, \alpha) = r > 0$, because every $c_v$ (for empty regions $v$) has the same distribution, we can create the full tree, and then uniformly choose the subset of $r$ regions which will have noisy count at least $\frac{10}{\epsilon} \cdot K \log T$. This takes time at most $T^{2K}$. We then have to compute each $c_v \sim \mathrm{Lap}(2/\epsilon)$ for the $r$ choices of $v$ where $c_v > \frac{10}{\epsilon} \cdot K \log T$. However, by using the well-known memoryless property of exponential distributions, the distribution of each $c_v$ is just $\frac{10}{\epsilon} \cdot K \log T + \mathrm{Expo}(2/\epsilon)$. Hence, we can compute the full data structure in time $O(T^{3K})$, in this setting. This, however, only happens with at most $T^{-4K}$ probability.

---

[4]The statement in [LN14] is for $\Phi = d^{O(1)}$, but applies to any $\Phi > 0$. The statement given here is by setting $R = d^{-1/q}$ in [LN14, Theorem 116] and scaling the minimal distance in the given metric to 1.

Alternatively, if $r = 0$, we can avoid computing any $c_v$ when $v$ is empty. The rest of the analysis in this case proceeds identically to subsection 3.3. Finally, in either case, the time to answer each query, in expectation, is still the same as in subsection 3.3.

Finally, we need to verify the accuracy of the algorithm. In expectation, each $c_v$ is off by at most $O(T \log K/\epsilon) = (\log n)^{O(1)}/\epsilon$, as the noise adds an expected $O(1/\epsilon)$ error, and the replacing of $c_v$ with 0 only happens when $c_v < O(T \log K/\epsilon)$. The analysis of Lemma 3.3 proceeds identically, except that the first source of error replaces the $\log \frac{1}{\delta}$ term with a $T \log K = (\log n)^{O(1)}$ term.

Finally, the embedding of arbitrary Euclidean data into the Euclidean sphere, as in Appendix A.2, proceeds the same way. Overall, Theorem 1.1 still holds when $\delta = 0$, except that the runtime of both the preprocessing and answering each query is in expectation, and the additive error removes the $\log(\log m/\delta)$ term. (Indeed, the additional $(\log n)^{O(1)}$ factors in the runtime and additive error can be absorbed into the $n^{o(1)}$ term).

**Large $\epsilon$.** We remark that our analysis extends to $\epsilon \leq n^{o(1)}$ in a straightfoward manner. We remark that in practice, because the probability ratio in differential privacy depends on $e^\epsilon$, it is highly impractical to use any $\epsilon$ significantly larger than $\Omega(\log n)$, and therefore we believe that it is not interesting to analyze an algorithm beyond $\epsilon \leq n^{o(1)}$.

To see why, note that for $1 \leq \epsilon \leq n^{o(1)}$, any $(1/2, \delta)$-DP algorithm is automatically $(\epsilon, \delta)$-DP. (This also holds in the $\delta = 0$ case.) Note that the runtime bounds do not depend on $\epsilon$, and the accuracy bounds are proportional to $1/\epsilon$ in the additive error. Therefore, if we allow the $n^{\rho+o(1)}$ term in the additive error to have a slightly larger $o(1)$ in the exponent (i.e., up to $n^{\rho+o(1)}/\epsilon$), the claim still holds until $\epsilon \leq n^{o(1)}$ for any $o(1)$.

**Adaptive Queries.** Here, we explain how we are able to improve Theorem 1.1 to handle adaptive queries.

We assume we are promised that all data points are in $[-R, R]^d$, where $R > r$ is some parameter. Let $\mathcal{T}$ represent the set of points $(x_1, \ldots, x_d)$ where every $|x_i| \leq 2R$ and $x_i$ is an integral multiple of $r/d$. Note that $|\mathcal{T}| \leq O(d \cdot R/r)^d = e^{O(d \log(dR/r))}$.

Now, consider having $M = O(d \log(dR/r))$ independent copies of the data structure, where each data structure is initialized with $(\frac{\epsilon}{M}, \frac{\delta}{M})$-DP. To answer each query, each data structure computes its approximation, and the overall data structure outputs the mean of the $M$ responses. By a standard Chernoff bound, and the accuracy bounds of Theorem 1.1, the output will be between $(1 - o(1)) \cdot |X \cap B_2(q, r)| - n^{\rho+o(1)} \cdot \frac{\log(\log m \cdot M/\delta) \cdot \log m}{\epsilon/M}$ and $(1+o(1)) \cdot |X \cap B_2(q, cr)| + n^{\rho+o(1)} \cdot \frac{\log(\log m \cdot M/\delta) \cdot \log m}{\epsilon/M}$, with $e^{-\Omega(M)}$ failure probability. In other words, the failure probability is $e^{-\Omega(M)}$ (assuming non-adaptive queries) and the additive error multiplies by at most $O(M \log M)$. Moreover, each data structure is $(\frac{\epsilon}{M}, \frac{\delta}{M})$-DP, so the overall algorithm, by basic composition, is $(\epsilon, \delta)$-DP. Finally, the runtime will only increase by a factor of $M$ (both for preprocessing and answering queries), since we have to repeat the process for $M$ data structures.

Now, we explain how we can deal with arbitrary adaptive queries. We can assume WLOG that every query is in the range $[-2R, 2R]^d$, or else the algorithm can simply output 0 without sacrificing either privacy or accuracy. Next, we can assume WLOG that every query is in $\mathcal{T}$, because any query $q$ is within $r/\sqrt{d}$ of some point $q' \in \mathcal{T}$, and answering the query on $q'$ will be accurate up to an additive error of $r/\sqrt{d} = o(r)$. Therefore, as long as the data structure is accurate for all of the queries in $\mathcal{T}$, this is sufficient.

Thus, for each $q \in \mathcal{T}$, the overall data structure will answer accurately with failure probability $e^{-\Omega(M)}$. For $M = O(d \log(dR/r))$, we have that $|\mathcal{T}| \leq e^{\Omega(M)}$, so by a union bound, the data structure will answer any query $q \in \mathcal{T}$ accurately. Therefore, even adaptive queries can be answered accurately, since even any adaptive query can be assumed WLOG to be in $\mathcal{T}$.

