# OpenReview forum: "Differentially Private Approximate Near Neighbor Counting in High Dimensions"
_NeurIPS.cc/2023/Conference — NeurIPS 2023 spotlight_

### Official Review · Reviewer_2zYg · 2023-07-04

**Soundness:** 4 excellent
**Presentation:** 3 good
**Contribution:** 3 good
**Rating:** 7
**Confidence:** 3

**Summary:**

This paper studies the problem of answering approximate range queries privately. The paper presents a data structure to answer queries privately that helps avoid dimension dependence in the additive error for utility but incurs a multiplicative factor. The paper also showcases how to efficiently implement the data structure.

**Strengths:**

The paper is well structured, easy to follow with adequate discussion and comparison with prior work.

The approach introduced provides a trade-off between additive error incurred and approximation factor for the query, which to the best of my knowledge is a novel contribution. The lower bound is for a restrictive case but understanding this dependence is a nice open problem.


**Weaknesses:**

The privacy guarantee is limited to $(\varepsilon, \delta)$-DP and not pure differential privacy. Some discussion on limitation or approach for extension to pure differential privacy might be helpful.

Also, it is not clear if the multiplicative factor in utility is necessary or inherent to this approach.

**Questions:**

Q1: Though $\varepsilon < 1$ is preferable for privacy, is there a barrier in extending Theorem 1.1 to case when $\varepsilon$ is $> 1$?

Q2: How does the bound in Theorem 1.1 compare to other approaches for the case of $c=1$?

**Limitations:**

Discussed in weakness section. Since work is of theoretical nature, negative societal impact is not apparent.

---

> ### Author Rebuttal · Authors · 2023-08-09
>
> W1: The privacy guarantee is limited to (ε,δ)-DP and not pure differential privacy. Some discussion needed.
>
> A: We could in fact obtain pure DP, although in this case the algorithm would suffer from a small probability of having large runtime. We focused on approximate DP for simplicity. To convert our algorithm into a pure DP algorithm, in lines 19-21 of the data structure, we add regular Laplace noise with parameter eps/2 (not truncated) to ensure pure DP, and replace c_v with 0 if it does not exceed roughly 1/eps * n^{o(1)} for some sufficiently large n^{o(1)} parameter. (The \Delta in line 20 is a typo and should be 1, we will fix( it.) The idea is that when the true count of a region is 0, its noisy count will not exceed the threshold with very high probability, but there is some small probability that it does exceed the threshold. We can compute the probability \zeta that none of the truly empty regions exceeds the threshold after adding noise by simulating a Binomial distribution. In the high probability event that this holds, we do not need to store any of the leaves. Otherwise, there is a small probability that we will store some (or even all) of the leaves and their noisy counts, which makes both the runtime and accuracy much worse. However, in expectation, the runtime and accuracy are still as stated. We will update the paper to discuss this more rigorously.
>
> W2: Not clear if the multiplicative factor in utility is necessary or inherent to this approach.
>
> A: The cardinality multiplicative factor is quite small (1+o(1)), but getting an algorithm that removes this factor (or lower bound showing its necessary) is an interesting open question.
>
> Q1: Though ε<1 is preferable for privacy, is there a barrier in extending Theorem 1.1 to case when ε is >1
>
> A: There does not seem to be any barrier, although for a somewhat technical reason. Specifically, if eps > 1, we can of course get the same bounds as when eps = 0.5. Furthermore, as long as eps = n^{o(1)}, we still get the same theorem, because in the expression n^{\rho+o(1)} the o(1) term becomes larger but remains o(1).  If eps is larger than n^{o(1)} the privacy guarantees are so weak that it is unclear whether the guarantees are meaningful. Still, the theorem might hold nevertheless. We will investigate this and discuss it in the final version of the paper.
>
> Q2: How does the bound in Theorem 1.1 compare to other approaches for the case of c=1
>
> A: There are generic techniques that work for arbitrary ranges (including Euclidean balls in high dimension) and achieve \sqrt{n} additive error, such as the paper [HR10] which we cite on line 124. In our setting, one can think of the Q and U terms being exponential in the dimension, though in our setting the dimension can be assumed as logarithmic in n by standard dimension reduction procedures. Hence, their bound is roughly proportional to sqrt{n}. To the best of our knowledge, no other results are known for high-dimensional Euclidean space.
>
> Our bound is only n for c = 1, as we mainly focused on getting the right dependence for larger approximation constants c. We will discuss the comparison of [HR10] with our work in more detail, and if it is possible to combine our techniques with [HR10] to get improved results for smaller c, we will include this in our final version of the paper.

---

> > ### Comment · Reviewer_2zYg · 2023-08-15
> > **Response**
> >
> > I thank the authors for their response. I believe including some discussions on these would make the results more complete and so I'm happy to increase my score.

---

### Official Review · Reviewer_1btN · 2023-07-05

**Soundness:** 3 good
**Presentation:** 2 fair
**Contribution:** 2 fair
**Rating:** 5
**Confidence:** 3

**Summary:**


The papers shows how to use a variant of LSH to approximately count the neighbors in an r-ball (r fixed apriori). It shows how to obtain a differentially private LSH sketch. It analyzes the algorithm’s theoretical properties.


**Strengths:**


The technical contributions look solid and the analyses seem correct, e.g. using a weaker Markov inequality to handle non-independence of points.

It shows how to avoid an exponential dependence on the dimension.

The algorithm itself is fairly clear.

The paper makes an interesting claim that their method also leads to the most space efficient approximate nearest neighbor search.


**Weaknesses:**

The presentation leaves a great deal to be desired. After the introduction, the exposition generally seems hastily arranged. Much of it is spent on the detailed analysis without offering much insight into either the algorithm or a high level overview of the proof strategy. The analysis skips details and crams equations into paragraphs which makes it harder to read. Since the proof is not short and many of the proofs are already in the appendix, it’d make sense to only outline the proof.

There is zero empirical validation. The result and algorithm are purely theoretical contributions.

There doesn’t seem to be any exposition supporting the claim that the paper yields the most space efficient known approx nearest neighbor search.

**Questions:**

1. The data structure created appears to be a (completely) random forest. What are the relationships between this work and differentially private random forests?

2. Where do the 0.9 probability using Markov’s and 99% probability using Chernoff’s come from? I don’t see any other constants chosen that ensure these probabilities are reached.

3. The extension to data not on a sphere is completely punted to the appendix. Can you give some insight on how this works, under what conditions it works, and what the impact to privacy is in the main text?

4. The overview says that you “force” the LSH algorithm to create proper partitions. But where do you do this? Is it from the tree structure with depth? (as opposed to a stump-like structure) And doesn’t a single projection in LSH partition points into buckets so you can do fast lookups? Or by partition do you mean some fine partitioning?

5.Can the structure answer a knn like query? I.e. What is the minimum radius r which contains k neighbors?


**Limitations:**

Yes

---

> ### Author Rebuttal · Authors · 2023-08-09
>
> W1: The presentation leaves a great deal to be desired.
>
> A: Apologies if the presentation was unclear. We will implement the reviewer’s suggestions in the final version of the paper.
>
> W3: There doesn’t seem to be any exposition supporting the claim that the paper yields the most space efficient known approx nearest neighbor search.
>
> A: We assume that the reviewer refers to the sentence “This is of separate interest, as this yields the most efficient algorithm for approximate nearest neighbor search with space O(nd), improving over [Kap15]”.  [Kap15] obtained a data structure with O(nd) space and query time dn^rho for rho=4/(c^2+1). Note that this gives a non-trivial query time only for c>\sqrt(3). In contrast, our data structure obtains query time with rho=4c^2/(c^2+1)^2 which is smaller than 1 for all c>1.
>
> Q1: The data structure created appears to be a (completely) random forest. What are the relationships between this work and differentially private random forests?
>
> A: Thank you for an interesting suggestion. We are not experts on differentially private random forests, but after a cursory examination of the literature we believe that random forests are typically data-dependent,  which makes it  hard to guarantee privacy for range queries. (For the same reason, in our paper we are using data-independent instead of data-dependent LSH, despite the fact that the latter has slightly better bounds.). If the reviewer has concrete suggestions for random forest papers we should compare our methods to, we will perform a more in-depth analysis.
>
> Q2: Where do the 0.9 probability using Markov’s and 99% probability using Chernoff’s come from? I don’t see any other constants chosen that ensure these probabilities are reached.
>
> A: We use 0.9 and 99% only to simplify the presentation. These probabilities can be made arbitrarily close to 1 by adjusting the big-Oh constant in the appropriate parameters (the error bound for Markov and the number of repetitions for Chernoff).
>
> Q3: The extension to data not on a sphere is completely punted to the appendix. Can you give some insight on how this works, under what conditions it works, and what the impact to privacy is in the main text?
>
> A: We moved this entirely to the appendix as it is essentially a corollary of Bartal, Recht, and Schulman [BRS11], as we note in lines 338-341. We convert solving the nearest neighbor problem on the sphere with radius r into solving the problem in Euclidean space (not on the sphere) with the same radius. The high level idea of the BRS11’s result is to provide an efficient map from Euclidean space onto the unit sphere, which ensures close points stay close and far points stay far. We use this result as a black box, and this reduces solving the problem on Euclidean space to solving it on the sphere, which is our main algorithm. There is a slight distortion (which BRS11 rigorously quantifies), but it is insignificant for our purposes. Because we perform the private computation after performing this embedding first, we do not sacrifice privacy.
>
> Q4: The overview says that you “force” the LSH algorithm to create proper partitions. But where do you do this?
>
> A:  The specific place in Algorithm 1 where this is done is line 17. Breaking out of the “for i” loop ensures that we map each point p to only one child of the node v.
>
> Q5: Can the structure answer a knn like query?
>
> We believe that given our data structure, one can use a similar approach to [HY21], and approximately retrieve the distance to the k-nn. In particular, given the parameter k, the goal is to retrieve r such that $r_k \leq r \leq (1+\alpha) r_{k(1+o(1))+2t}$ where $r_i$ is used to denote the smallest radius around $q$ such that the ball $B(q,r_i)$ contains at least $i$ points. The approach of [HY21] is to roughly perform a binary search to approximately find the right radius at which there are at least k points. Please refer to Lemma 13 and Theorem 14 of [HY'21].
>
> We note, however, that there are two differences in our case:
>
> *  [HY21] assumes that the data points are from the discrete cube [u]^d and thus the minimum and maximum radius are bounded by 1 and $(\sqrt d)u$. Our algorithm does not have this restriction and therefore, in order to answer k-nn type queries, one needs to further assume a bounded aspect ratio. Given that the minimum distance to any point is 1 and the maximum distance is $\sqrt d u$, the algorithm of [HY21] queries the data structure $g = log_{1+\alpha} (\sqrt d u)$ many times to find the right radius at which there are at least k points included in the data structure.
>
> * However in our case, our algorithm works for a fixed r, and thus we now need to maintain g different data structures. Thus we need to use stronger privacy parameters in each data structure so that using composition theorems the overall privacy is preserved. This is as opposed to [HY21] where they did not need a separate data structure for different values of r and could reuse their data structure to answer queries for all values of r.
>
> We will verify this, and if correct, include this observation in the final version.

---

> > ### Comment · Reviewer_1btN · 2023-08-14
> >
> > Thank you for your rebuttal. I've bumped up the rating.

---

### Official Review · Reviewer_3KJE · 2023-07-06

**Soundness:** 4 excellent
**Presentation:** 4 excellent
**Contribution:** 3 good
**Rating:** 7
**Confidence:** 3

**Summary:**

This paper provides a new polytime algorithm for differentially private approximate near neighbor counting---that is, privately counting the number of points inside l2 balls of fixed radius r, or more precisely a relaxation that may answer any value between the number of datapoints in B(x,r) and the number of datapoints in B(x, c*r) for parameter c.

Existing algorithms incur either additive error that is at least a fixed polynomial in the number of points n or additive error that is logarithmic in n but grows exponentially in the dimension d. The algorithm in this paper incurs additive error that is n^(O(1/c^2)) where c is the parameter above, which can be made an arbitrarily small polynomial in n and has no dependence on d, at the cost of also introducing a small multiplicative error term.

(As a side result, the paper also provides a lower bound showing that for L-infinity balls, it is not possible to have additive error n^(o(1)) for constant c. It poses the more immediately relevant lower bound question for L2 balls as an open question.)

**Strengths:**

This is a nice result that improves on the additive accuracy of known results when both n and d are large. The algorithm is natural and interesting, and the paper is well-written and enjoyable.

**Weaknesses:**

It's not clear whether the multiplicative approximation factor is necessary to get a good additive dependence on both n and d. (It's perfectly reasonable to trade off a small multiplicative factor for a better additive term, so this is a bit of a nitpick. However, it does mean that the result is incomparable rather than strictly better compared to prior work even in the regime when n and d are both large.)

Relatedly, the lower bound (in addition to being for l_\infty rather than l_2) does not take into account the multiplicative approximation, so it's not clear whether it's possible to obtain additive error n^{o(1)} for constant c if we also allow a multiplicative error.

Minor comments:
The abstract should mention the multiplicative aspect of the approximation.
42: Maybe say explicitly \ell_2 ball here, not just ball.
69: The accuracy guarantee should only hold with high probability for the m queries q_1, ..., q_m, not for every q\in \mathbb{R}^d.
128: I'm not positive, but I think one of the interval query results of BNSV15 has been improved in subsequent work to reduce the gap between the upper and lower bounds.
200: It's technically not a partition, since some points are not sent to any child.

**Questions:**

1) Does the algorithm extend to other norm balls or other natural families of queries?

2) Is there hope of removing the multiplicative approximation factor? Or else reason to believe that it is necessary?

3) Do you think the analysis could go through for adaptive queries as well, or is it clear that the restriction to non-adaptive queries is necessary?

**Limitations:**

Yes.

---

> ### Author Rebuttal · Authors · 2023-08-09
>
> W1: It's not clear whether the multiplicative approximation factor is necessary to get a good additive dependence on both n and d. ... Relatedly, the lower bound (in addition to being for l_\infty rather than l_2) does not take into account the multiplicative approximation.
>
> A: Indeed, we do not know the answer to this question.  The cardinality multiplicative factor is quite small (1+o(1)), but getting an algorithm that removes this factor (or lower bound showing it is necessary) is an interesting open question.
>
>
> Q1: Does the algorithm extend to other norm balls or other natural families of queries?
>
> A: The algorithm should extend easily to L1 balls, with somewhat worse performance bounds. This is because the L1 norm can be embedded into the L_2 norm squared, which (for range queries) is equivalent to the L_2 norm, though the approximation factor c becomes c^2. We will verify this, and if correct, include this observation in the final version.
>
>
> Q3: Do you think the analysis could go through for adaptive queries as well, or is it clear that the restriction to non-adaptive queries is necessary?
>
> A: We believe that it should be possible to extend the analysis to adaptive queries, at the cost of multiplying the additive error and runtime by a factor roughly d. The idea is that it should be possible to round every query point to an r-net, and then we just need an algorithm able to answer any query among (1/\r)^O(d) queries accurately with failure probability (1/\r)^{-O(d)}, so that by a union bound all queries are answered correctly. Such a data structure can be constructed by replicating the data structure in the paper d log(1/r) times. Since we need many replicas, each copy needs privacy with parameter eps replaced by eps/d, which multiplies the error/runtime by a factor of poly(d). We will verify this, and if correct, include this observation in the final version.

---

### Official Review · Reviewer_TS42 · 2023-07-07

**Soundness:** 4 excellent
**Presentation:** 4 excellent
**Contribution:** 3 good
**Rating:** 7
**Confidence:** 4

**Summary:**

In this work, the authors propose a differentially private data structure to approximately count the number of data points from within a dataset that lie within a certain small radius of a query point.

The preliminary data structure, intended for datasets that lie on a unit sphere, recursively splits the region into small caps. This is represented using a T-ary tree of height K. The leaf nodes of this tree maintain counts of all the points that lie within their corresponding regions. Privacy is obtained by perturbing these points using a truncated Laplacian mechanism.

The proofs follow from simple Gaussian concentration bounds and standard Laplacian mechanism.

**Strengths:**

- The problem is relevant and interesting.
- The solution is simple and elegant and circumvents the drawbacks of prior works.
- The authors provide a novel insight for using Locality sensitive hashing schemes for approximate counting.
- The presentation is very clear and concise.

**Weaknesses:**

- The improved results only hold for small values of r which is fixed in advance and is not a part of the input.

**Questions:**

- Can you convert any data-independent LSH with LSH constant \rho to construct an approximate near neighbor counting using a similar tree structure in a black-box fashion? For instance, each internal node will correspond to all the points from the parent node that fall within a certain region. The error estimates will be analyzed in a similar fashion and will be guaranteed by the LSH parameters.
The insight comes from the \rho used in the theorem statement whose value is exactly the optimal LSH constant for data-independent schemes.

- How about data-dependent LSH? The construction of Andoni & Razenshteyn provides a better LSH constant and does a recursive splitting that looks similar to the one used in this work.

**Limitations:**

Yes.

---

> ### Author Rebuttal · Authors · 2023-08-09
>
> W: The improved results only hold for small values of r which is fixed in advance and is not a part of the input.
>
> A: It is true that our algorithm assumes that r is fixed. However, it does not need an assumption that r is small, see Appendix A.2.
>
>
> Q1: Can you convert any data-independent LSH with LSH constant \rho to construct an approximate near neighbor counting using a similar tree structure in a black-box fashion? For instance, each internal node will correspond to all the points from the parent node that fall within a certain region. The error estimates will be analyzed in a similar fashion and will be guaranteed by the LSH parameters. The insight comes from the \rho used in the theorem statement whose value is exactly the optimal LSH constant for data-independent schemes.
>
> A: The answer depends on whether privacy is considered. Without the privacy constraint, it is indeed possible to transform any LSH scheme for ANN search into ANN counting data structure. But once we consider the privacy constraint, there does not seem to be a natural way to do so. Specifically, our algorithm builds a decision tree that partitions that space/dataset into buckets, where the query looks up a subset of buckets. Constructing such a structure for an arbitrary LSH family is, to our knowledge, an open problem.
>
>
> Q2: How about data-dependent LSH? The construction of Andoni & Razenshteyn provides a better LSH constant and does a recursive splitting that looks similar to the one used in this work.
>
> A: Even though data-dependent LSH indeed gives better exponents, it seems particularly challenging to leverage it in a differentially private algorithm.  One challenge is that the space partitions intrinsically depend on the dataset, and hence it may be hard to control the privacy leakage here. Another challenge is that all known data-dependent LSH schemes are not pure space partitions --- a crucial condition we use to obtain the 1+o(1) multiplicative approximation factor --- and would instead give a super-constant factor.

---

> > ### Comment · Reviewer_TS42 · 2023-08-15
> >
> > Thanks for entertaining my questions. I enjoyed reading the paper.

---

### Author Rebuttal · Authors · 2023-08-09

We thank all reviewers for their useful comments and feedback. We will fix the typos and presentation issues in the final version of the paper. In what follows we address the issues identified by the reviewers as weaknesses and/or listed as questions.

---

### Decision · Program_Chairs · 2023-09-21

**Decision:**

Accept (spotlight)

**Comment:**

This paper studies approximate Near neighbor counting with differential privacy. The reviewers all agreed that the paper is interesting, and presents a natural and elegant solution. Any reviewer concerns were largely addressed in the discussion phase, and I would encourage the authors to update the paper based on the feedback, and the rebuttal. I am happy to recommend acceptance.